# Malaysia’s Health Systems Response to COVID-19

**DOI:** 10.3390/ijerph182111109

**Published:** 2021-10-22

**Authors:** Zen Yang Ang, Kit Yee Cheah, Md. Sharif Shakirah, Weng Hong Fun, Jailani Anis-Syakira, Yuke-Lin Kong, Sondi Sararaks

**Affiliations:** 1Institute for Health Systems Research, Ministry of Health Malaysia, Shah Alam 40170, Selangor, Malaysia; shakirah.ms@moh.gov.my (M.S.S.); fun.wh@moh.gov.my (W.H.F.); anissyakira.j@moh.gov.my (J.A.-S.); kong.yl@moh.gov.my (Y.-L.K.); sararaks.s@moh.gov.my (S.S.); 2Institute for Clinical Research, Ministry of Health Malaysia, Shah Alam 40170, Selangor, Malaysia; cheahkityee@crc.moh.gov.my

**Keywords:** response, measures, strategies, public health, Malaysia, COVID-19, Movement Control Order, lockdown, pandemic, health systems

## Abstract

This study aimed to highlight the COVID-19 response by the Ministry of Health (MOH) and the Government of Malaysia in order to share Malaysia’s lessons and to improve future pandemic preparedness. The team conducted a rapid review using publicly available information from MOH, PubMed, and World Health Organisation (WHO) Global Research on Coronavirus Disease Database to compile Malaysia’s responses during the COVID-19 pandemic. Measures taken between 31 December 2019 and 3 June 2020 were classified into domains as well as the pillars described in the WHO COVID-19 Strategic Preparedness and Response Plan (WHO SPRP). Malaysia’s response incorporated all pillars in the WHO SPRP and consisted of five domains, (i) whole-of-government, (ii) cordon sanitaire/lockdown, (iii) equity of access to services and supports, (iv) quarantine and isolation systems, and (v) legislation and enforcement. Some crucial measures taken were activation of a centralised multi-ministerial coordination council where MOH acted as an advisor, with collaboration from non-government organisations and private sectors which enabled an effective targeted screening approach, provision of subsidised COVID-19 treatment and screening, isolation or quarantine of all confirmed cases, close contacts and persons under investigation, with all strategies applied irrespective of citizenship. This was provided for by way of the Prevention and Control of Infectious Diseases Act 1988. A combination of these measures enabled the nation to contain the COVID-19 outbreak by the end of June 2020.

## 1. Introduction

In December 2019, a cluster of respiratory disease cases of an unknown cause was detected in Wuhan, Hubei Province, which was subsequently attributed to the severe acute respiratory syndrome coronavirus 2 (SARS-CoV-2) [1]. The disease spread quickly within China and across the globe where Thailand, a country bordering Malaysia, was the first country outside of China to record its first confirmed case merely two weeks after the outbreak was declared [1]. The disease reached Malaysia on 25 January 2020 with three positive cases reported [1,2,3]. The rapid spread of the disease globally prompted the World Health Organisation (WHO) to declare the outbreak a pandemic on 12 March 2020.

With a basic reproduction number (R0) between 2 and 3 [4], the SARS-CoV2 disease spread swiftly between people who were in close contact. As the pandemic progressed, without any effective prevention and cure, countries around the world have attempted to break the chain of infection by employing public health measures. WHO released the Coronavirus Disease 2019 (COVID-19) Strategic Preparedness and Response Plan (WHO SPRP) in early February 2020, focusing on eight pillars encompassing public health areas central to preparedness and response activities, with the ninth pillar added in May 2020 [5,6]. The nine pillars were, namely,

Pillar 1: Country-level coordination, planning, and monitoring.

Pillar 2: Risk communication and community engagement.

Pillar 3: Surveillance, rapid response teams, and case investigation.

Pillar 4: Points of entry, international travel, and transport.

Pillar 5: National laboratories.

Pillar 6: Infection prevention and control.

Pillar 7: Case management.

Pillar 8: Operational support and logistics.

Pillar 9: Maintaining essential health services and systems.

Acknowledging existing national preparedness and response plans in member states, WHO advised adaptation of the WHO SPRP into existing country-specific preparedness and response plans before 30 April 2020 [5,6].

In anticipation of COVID-19’s possible transmission to Malaysia, the Ministry of Health (MOH) Malaysia implemented pre-emptive measures to ensure preparedness and subsequently manage the nation’s COVID-19 pandemic [7,8]. On two occasions (16 March 2020 and 11 May 2020), WHO had rated Malaysia at the highest level, Level 5 (more than 80% of benchmark capacity level) in terms of country preparedness capacity based on WHO Country Preparedness and Response Status for COVID-19 [9,10,11]. Nevertheless, to improve the country’s future preparedness and response for experiential learning [12] as well as to share lessons learnt with the world, this study aimed to review the COVID-19 pandemic response by the MOH and the Government of Malaysia.

## 2. Materials and Methods

A document review using rapid review methodology [13] was performed to identify publicly available information on the measures employed in Malaysia to address the COVID-19 pandemic. MOH-affiliated official websites were first searched for relevant information/documents containing COVID-19 measures/strategies and case occurrence in Malaysia. Besides, WHO global research on the coronavirus disease database (WHO COVID-19 database) [14] and PubMed (https://pubmed.ncbi.nlm.nih.gov/ (accessed on 26 August 2020) were searched for relevant literature (refer to Appendix B).

The MOH websites search was completed between 27 March and 18 April with an updated search on 1 July to 5 July 2020. Publicly available documents in English or Malay from 31 December 2019 to 30 June 2020 were included. WHO COVID-19 database was searched for relevant articles on 24 March 2020 (with an updated search on 18 April and 5 July 2020) using the terms as listed in Appendix B. The PubMed database was searched using the same terms. Articles (including preprints) in English describing Malaysia’s response to the COVID-19 pandemic between 31 December 2019 and 30 June 2020 were included. An additional supplemental search for local and international key events and/or measures from other Malaysian government and private organisation websites as well as local newspapers (refer Appendix B) was performed to contextualise the findings from MOH websites, PubMed, and WHO COVID-19 databases.

Articles and documents that were included in the final analysis had to fulfil the inclusion criteria of containing any health systems measure that were performed to tackle the COVID-19 pandemic in Malaysia (refer to Appendix B). Journal articles (including preprints) or documents that did not discuss any health system measure or strategy were excluded. The document search and selection were completed according to Preferred Reporting Items for Systematic Reviews and Meta-Analysis (PRISMA) [15].

Title and abstract screening were undertaken by one reviewer, and verification of inclusion or exclusion was completed on 10% of documents by another reviewer following the methodology by Tricco et al. [16]. Reviewers independently continued to screen documents after >90% agreement was observed. Any discrepancies were resolved through discussion [13]. Measures described in Malay were translated into English. Extraction of measures from included documents was completed in Microsoft Excel by a reviewer, with all entries verified by another reviewer. Disagreements were resolved by discussion with four other members.

The extracted measures were initially classified according to the pillars described in WHO SPRP. However, as the team members found that a single measure employed by Malaysia could be categorised under several pillars concurrently, categorising directly according to the pillars caused duplication of strategies. Hence, to classify Malaysia’s response to COVID-19, the extracted measures were first categorised into subdomains before grouping into domains and then subsequently classified to the nine pillars described in WHO SPRP [5,6]. The sub-domains were formed by analysis of content extracted. Group discussions were carried out to reach consensus. These were carried out independently by reviewers, with all categorisation of measures into sub-domains, domains, and the corresponding pillars verified by another reviewer. These measures, contextualised by local and international key events, were presented according to phases of the outbreak (i.e., before the first wave, during the first wave and the second wave) and COVID-19 reported cumulative confirmed cases, active cases, deaths, and daily new cases in Malaysia to illustrate the chronology of events and case occurrence in the country. The nine pillars provided a framework for researchers to systematically analyse and synthesise the country’s response to COVID-19.

## 3. Results

We retrieved 248 documents from MOH websites, 52 articles from both PubMed (*n* = 43) and WHO COVID-19 databases (*n* = 9) as well as 106 documents from supplemental search through other sources, respectively. Of these, 233 documents from MOH-affiliated websites, 10 articles from PubMed/WHO COVID-19 databases, and 103 documents from supplemental search met the inclusion criteria (Figure 1, Appendix A; Appendix C).

The crucial events, health systems measures taken by Malaysia and COVID-19 cases are illustrated in Figure 2. The number of daily reported cases in Malaysia started to drop two weeks after implementation of the nationwide cordon sanitaire (i.e., “the closing of local or community borders or erection of a barrier around a geographic area with strict enforcement to prohibit movement into and out of the area” [17]), namely the Restriction of Movement Order (locally known as Movement Control Order, MCO) on 18 March 2020 (Figure 2, Appendix A). Malaysia’s key responses in the pandemic consisted of five domains, namely (i) whole-of-government and whole-of-society approach, (ii) cordon sanitaire or lockdown, (iii) equity of access to services and supports, (iv) quarantine and isolation systems, and (v) legislation and enforcement. The nation’s involvement and contributions to surge capacity, surveillance activities, delivery of medical services as well as COVID-19 related physical and digital infrastructure such as mobile applications for COVID-19 contact tracing and updates, were achieved by a centralised coordination council (i.e., National Security Council, NSC) consisting of multiple ministries where MOH provided advice based on the evolving COVID-19 situation, as well as collaborations with other countries, non-government organisations, private sectors, and the public (Figure 3 and Appendix D).

On 9 March 2020, Brunei International Health Regulations Focal Point informed MOH Malaysia that one Brunei citizen tested positive for COVID-19 after returning from an annual religious gathering involving more than 15,000 congregants from Malaysia and neighbouring countries held at Sri Petaling, Selangor, Malaysia, from 27 February until 1 March 2020 (Figure 2) [18,19,20,21,22]. Those participants who unknowingly contracted COVID-19 during the religious gathering returned to their hometowns or religious schools across Malaysia after leaving the religious gathering, causing a sudden upsurge of COVID-19 positive cases in Malaysia [21,22]. Consequently, Malaysia implemented a countrywide partial lockdown (i.e., MCO) to contain the outbreak. After the nation had fulfilled the WHO criteria for lifting the lockdown, the nationwide partial lockdown was relaxed and instead, the country transitioned into the implementation of Conditional MCO (CMCO) and Recovery MCO (RMCO) (Figure 2, Table 1 and Appendix E). In addition, for localities with a sudden surge of cases, localised completed lockdown with mass screening (Enhanced MCO, EMCO) or partial lockdown with mass screening (Administrative MCO, AMCO) were implemented. During the lockdown period, mass gatherings were prohibited, with points of entries and socioeconomic activities regulated by standard operating procedures (SOPs) issued by the NSC (Table 1 and Appendix E).

Furthermore, Malaysia adopted a targeted screening approach for case detection and monitoring, where all individuals identified as high-risk, such as incoming travellers and individuals from EMCO and AMCO areas as well as religious schools (due to a possibility of teachers and/or students having attended the aforementioned Sri Petaling Religious Gathering) [22], were screened and sampled if they fulfilled the suspected case definition (Figure 4). The case detection capacity of targeted screening was enhanced with the involvement of multiple agencies in the process, such as the police, United Nations High Commissioner for Refugees, and Department of Islamic Development Malaysia (Figure 3 and Appendix D). For example, EMCO was enforced in Simpang Renggam (in Johor) after 79 COVID-19 cases were detected in the region within a short period of time [23]. During the implementation of the EMCO in Simpang Renggam region, which consisted of several townships and villages that housed 3755 individuals, 1634 (45%) residents had been actively traced, screened, and sampled with an additional 59 positive cases detected within a period of one week [23,24]. With these efforts, as of 19 June 2020, the targeted screening approach was able to detect the majority (7314 cases, 86%) of total confirmed cases (8353 cases) in the country (Figure 4) [25]. Meanwhile, as of 30 June, a total of 765,000 individuals in the nation were sampled (i.e., cumulative population test rate of 23.4 individuals tested per 1000 population), with a positive rate of 1.13% (8639 cases) [26].

Meanwhile, hospital and non-hospital quarantine systems were designated to manage COVID-19 cases according to the degree of disease severity and risk of infection, so that medical facilities and resources can be appropriately channelled (Table 2 and Appendix F). Equity of access to various services and supports including screening, testing, subsidised treatment, personal protection equipment (PPE), essential services, food supply, and financial assistance was provided regardless of citizenship pursuant to the Fees (Medical) (Cost of Services) Order 2014 and the Prevention and Control of Infectious Diseases Act 1988 as non-citizens would otherwise be charged with the full cost. COVID-19 information was also provided promptly to the public through conveying official daily COVID-19 updates using various communication channels, engagement of religious leaders to provide advice and updates, as well as the publication of guidelines related to COVID-19 management in different contexts (e.g., guidelines for the management of dead bodies and custodial settings) (Figure 5 and Appendix G). Lastly, relevant laws and regulations that were enforced during the COVID-19 pandemic enabled the government to handle the public health crisis with public compliance (Figure 6 and Appendix H). Responses were intensified during the second wave of the outbreak (27 February onwards) which corresponded with the increase of total active cases and daily new cases during this period (refer to Appendix D, Appendix E, Appendix F, Appendix G, Appendix H and Appendix I; Appendix A).

Matching of WHO Pillars with Malaysia’s responses demonstrated that all major public health measures outlined by WHO SPRP were addressed. The domains typically straddled several pillars rather than being unique to one pillar (Figure 3, Figure 5 and Figure 6, Table 1 and Table 2). The nation saw a reduction in daily new cases and active cases by the end of June (Figure 2).

## 4. Discussion

### 4.1. Principle Findings

Through whole nation collaboration, Malaysia enforced lockdowns with different restrictions to contain the COVID-19 outbreak, adopted a targeted screening approach to screen and detect COVID-19 cases, implemented institutionalised isolation and quarantine systems to house all confirmed cases, persons under investigation (PUIs), and close contacts at gazetted facilities as well as provided equal care to both citizens and non-citizens. These public health strategies not only addressed all the recommendations in the WHO SPRP, but also successfully contained the outbreak and allowed socioeconomic activities simultaneously.

### 4.2. Implications to the Stakeholders, Lessons Learnt, and Future Direction

#### 4.2.1. Lockdown—The Rationale and Impact?

Several countries have implemented various degrees of lockdown since the COVID-19 outbreak [27,28,29,30]. Even though a lockdown implementation has reduced the transmissibility of COVID-19 in several countries, it also caused collateral damage, such as negative psychosocial effects [31,32,33,34], increase in domestic violence [35], reduction in physical activities and economic impact [36].

Malaysia implemented a nationwide partial lockdown (MCO) (refer to Table 1) to decrease the disease transmission rate which in turn helped to reduce the burden to the country’s health system. The possible reasons of such a nationwide measure were the increasing number of active cases in March (Figure 2) as well as the fact that the Sri Petaling religious congregants had returned to their hometowns or religious schools across Malaysia after participating in the religious gathering, resulting in the disease spreading countrywide [22,37]. The nationwide lockdown had restricted the movement of the public and high-risk groups (e.g., Sri Petaling religious congregants) and therefore prevented further spreading of COVID-19 while enabling more time for detection and isolation of cases using the targeted screening approach [22,37]. While the lockdown managed to contain the outbreak and reduce the number of new cases, it also affected public livelihoods [21,38]. To resolve this, the government allowed some economic sectors to re-open in stages subject to conditions such as compliance to standard operating procedures (SOPs) issued by National Security Council (NSC) and provided economic stimulus packages to ease the burden on affected industries and individuals (Table 1 and Figure 5) [21,38,39]. These countermeasures were employed to minimise disease transmission and safeguard the country’s economy simultaneously during the nationwide partial lockdown.

In later stages of the pandemic, EMCO and AMCO were implemented in targeted localities with a sudden increase of COVID-19 cases. The possible rationales for these localised lockdowns were as follows: (i) outbreaks were localised rather than widespread like the Sri Petaling Religious Gathering Cluster [20], (ii) implementation of comprehensive contact tracing systems (using mobile applications) that increased traceability of COVID-19 cases, and (iii) implementation of SOPs for infection and prevention control (IPC). EMCO and AMCO allowed MOH to contain the outbreak by carrying out active case detection in affected areas, whilst enabling continued livelihoods outside EMCO and AMCO areas. Similarly, relaxation of lockdown (CMCO and RMCO) allowed most of the socioeconomic activities to resume in stages, subject to conditions and standard operating procedures (SOPs) to prevent another COVID-19 outbreak in the community (Table 1).

In short, different types of lockdowns were enforced in Malaysia in response to COVID-19 with the aim to target different epidemiological scenarios that were happening as well as based on experiential learning.

#### 4.2.2. Mass Gathering—The Importance of Regulating a Mass Gathering in the Future

The rise in the number of cases arising from a religious congregation both locally [40] and in countries such as Korea [41] highlighted the importance of regulating mass gatherings in preventing the spread of COVID-19. From this large cluster of cases, mass gatherings were prohibited in the country, standard operating procedures (SOPs) were enforced, and preventive public health measures were emphasised. To ensure that people in Malaysia continue to practice IPC measures, the Compliance Operations Task Force was established to enforce the SOPs during the relaxation phases of lockdown, which were CMCO and RMCO.

#### 4.2.3. COVID-19 Surveillance—Expand the Eligibility for COVID-19 Testing and Conduct Mass Testing?

A targeted screening approach, such as that adopted by MOH Malaysia, screens, tests, and then isolates specific populations that were identified of having a higher risk of contracting COVID-19 as well as traces all close contacts to these cases [42,43,44]. WHO recommends that countries with clusters of COVID-19, such as Malaysia, test all suspected cases rather than carrying out mass testing [45,46]. Currently, Malaysia’s COVID-19 surveillance conforms to the recommendations by WHO. Additionally, Malaysia took an extra step by performing mass screening of high-risk groups and at EMCO and AMCO locations to detect suspected cases, i.e., targeted screening approach, which achieved a cumulative population testing rate of 23.4 individuals per 1000 population as of 30 June [26].

On the contrary, mass testing or population-wide testing is defined as a large scale COVID-19 laboratory testing on individuals with or without COVID-19 symptoms in a given population which aims to detect and quarantine every individual with active infection [47]. This is followed by contact tracing to monitor outbreaks in the community [48,49,50]. Mass testing on a population after detection of the first case in a community (e.g., Municipality of Vo’, Italy) [51] or testing of all patients with symptoms of influenza-like illness (ILI) (e.g., Victoria State, Australia) was carried out because it allowed detection of sporadic, asymptomatic, or pre-symptomatic cases in the community earlier and in turn prevented further outbreak resulting from unforeseen events such as the wave of cases after the religious congregation in Kuala Lumpur. A study done by the European Centre for Disease Prevention and Control reported that 9 out of 26 countries included in the study were conducting various degrees of mass testing [47]. For example, the State Government of Victoria, Australia and the United Kingdom expanded the eligibility of free COVID-19 testing for everyone who presented with ILI symptoms [48,52], and the Government of Hong Kong not only provided free testing for other high-risk groups such as pregnant women and front liners but also conducted the voluntary Universal Community Testing Programme for every individual in Hong Kong [53,54]. For example, surveillance data from the United Kingdom indicated that the Pillar 2 COVID-19 testing programme which included a wider population through commercial partnerships, had conducted 12.6 million COVID-19 tests and detected 202,496 positive cases as of 27 September [55,56,57]. The mass testing involving 10 million individuals in Wuhan in June 2020 found 300 asymptomatic positive cases and the results of all close contacts (*n* = 1174) of these positive cases were negative [58]. Similarly, the voluntary Universal Community Testing Programme carried out in Hong Kong from 1 to 14 September 2020, which collected 1.8 million samples for COVID-19 nucleic acid test, found that 13 out of 32 new positive cases were asymptomatic [54,59]. From these examples, it can be seen that mass screening can detect sporadic and asymptomatic cases in the community.

Nevertheless, due to limited evidence, the role of asymptomatic or pre-symptomatic transmission remains unclear [60]. According to Malaysia’s Director-General of Health, MOH Malaysia did not adopt mass testing as it might not be effective as an individual could still be exposed to COVID-19 infection post-testing and thus the optimal interval of testing was not known. It was therefore not cost-effective [61] and could instil a false sense of security among those who tested negative [62]. The study by European Centre for Disease Prevention and Control concluded that mass testing could be considered when a country intends to rapidly reduce the transmission within the community or when a country with very low levels of transmission intends to eliminate COVID-19 [47]. Nevertheless, if a mass testing programme is to be launched, it should be based on up-to-date evidence as it requires a clear definition of the population’s testing eligibility, well-coordinated health systems, and equitable access [49]. Without these elements, the programme would not be cost-effective [49].

#### 4.2.4. Adopting Digital Technology in Fighting the COVID-19 Pandemic

The legal framework and infrastructure in South Korea allowed the country to swiftly utilise closed-circuit television footage, medical facility records, phone-based global positioning systems, and credit card transactions for ease of epidemiological investigation and contact tracing for citizens or visitors in Korea as early as mid-February, three weeks after detection of the index case in South Korea [63,64]. The country also required individuals under self-quarantine to install the “self-quarantine safety protection mobile application” and any violation of self-quarantine order could be traced using GIS surveillance dashboard [64,65]. In contrast, the Malaysian government’s initiative in using technology for contact tracing, surveillance, and quarantine purposes were not observed initially. Various technology innovations were only introduced from April 2020 onwards [66,67,68].

However, as these mobile applications were developed by different agencies, their functions differed, with separate databases. This could confuse the users especially those who are not information technology savvy. China faced similar issues when provinces developed their regional Quick Response (QR)-code based mobile applications in the beginning. Nevertheless, on 18 March 2020, the Central Government of China required nationwide merging of all mobile applications by indicating three approaches to follow [69]. Similarly, the Federal Government of Malaysia and the Selangor State Government initiated a collaboration to integrate the databases of MySejahtera and SELangkah mobile applications [70] and subsequently, PgCare mobile application ceased its services after 31 August 2020 to enable users in Penang State to use MySejahtera mobile application only [71]. This allows the public to access multiple functions by using a single application. Hence, more organised data could be collected digitally.

Additionally, the Malaysian government could consider the Korean government strategy, which is adding on a reporting function into its existing contact tracing mobile application [65]. This enables the public to play a role in assisting enforcement by reporting individuals who violate regulations.

#### 4.2.5. Information and Risk Communication—Addressing Infodemic Needs

Since the COVID-19 pandemic began, MOH and other relevant agencies in Malaysia such as NSC and Malaysian Communications and Multimedia Commission utilised both conventional and digital media to communicate with the public [72]. Despite these measures, rumours and fake news were still circulated privately in social media such as WhatsApp or Facebook posts that required clarification from MOH and these agencies.

This indicates that risk communication, including the presentation of epidemiological data, could be improved. Singapore and Hong Kong reported details of specific locations (i.e., building) of each COVID-19 confirmed case [73,74] while Malaysia provided information of cases within a 1 km radius using the MySejahtera application. As rumours and viral news were mainly on the location of new cases, a future strategy could be to provide more precise location details to the public. This could in turn reduce the occurrence of fake news as well. Perhaps, Malaysia could benefit by developing an official COVID-19 dashboard with geographic information systems [75,76] to illustrate more precise locations of COVID-19 cases so that the public can access accurate updated information easily and the local residents could be more vigilant. Nonetheless, precaution should be exercised when revealing information such as the location of COVID-19 cases as the action might cause social stigmatisation towards the patients, organisations, and local residents involved [77]. With the consistent and standardised presentation of information to the public from the beginning, misinformation and confusion could be avoided and therefore the public confidence in the government’s handling of the pandemic could be strengthened.

#### 4.2.6. Outbreak Containment in Custodial Settings

MOH Malaysia regularly published and updated guidelines, including those for custodial settings, since March 2020 [78]. Akiyama et al. [79] and Kinner et al. [80] reported that the risk of an outbreak is higher in custodial settings. Malaysia initially recorded outbreaks in several immigration detention centres at the end of May 2020 (Figure 2) [81]. However, at the time of writing this discussion, another outbreak was reported in September originating from a lockup that detained two illegal immigrants resulting in more than 1400 individuals being screened [82]. The cause of these outbreaks might be due to the detainee’s rRT-PCR tests showing false negative as the virus was in the incubation period [83] which was documented by Lauren et al. [84]. Analysis by MOH Malaysia found that overcrowded and confined conditions in these settings were the primary factors for rapid disease spread, a condition well documented in current literature [79,80].

Therefore, to prevent a future outbreak in these settings, measures such as constantly strengthening SOPs implementation, quarantine of new inmates before mixing with existing inmates, conducting a second COVID-19 test for a new inmate after the first negative test, improving overcrowding in custodial settings, reducing the interfacility transfer of inmates, limiting visitations by attorney, and suspension of family members visits or offering videoconferencing services instead of physical visits, should be considered [79]. Moreover, multiple agencies collaboration such as cooperation from courts to accelerate sentencing procedures especially for minor crimes to reduce overcrowding these centres [79] and tightening of land border control to reduce the entry of illegal immigrants (such as in Ops Benteng) [85] could be intensified. Nevertheless, all detainees, prisoners, non-citizens including illegal immigrants and refugees in Malaysia were given equal care regardless of nationality and social status [86].

#### 4.2.7. Importance of Developing Guidelines Suited to Local Culture and Context

Bruns et al. described that certain cultural and religious practices might increase the risk of COVID-19 transmission and emphasised the importance of adopting new norms in these practices [87]. A South African study showed the importance of compliance to IPC measures such as social distancing, personal hygiene and donning a face mask as the majority of COVID-19 cases in Eastern Cape Province in South Africa was linked to burial ceremonies and religious services due to lack of compliance to these IPC measures [88]. In a multiracial nation like Malaysia, understanding diversity, mutual respect, and cultural competence are essential in producing culturally sensitive guidelines. Hence, MOH Malaysia COVID-19 management guidelines were adapted to suit the country’s unique culture and need. Additionally, Malaysia ensured good compliance to SOPs during religious events (e.g., burial ceremony) by engaging with religious bodies and providing professional advice to these organisations before publishing the guidelines of handling of dead bodies of COVID-19 cases (Figure 5 and Appendix G) [23,89]. This conforms with the recommendations by WHO [90].

### 4.3. Strengths and Weaknesses

Firstly, this review compiled strategies extracted from selected Malaysian government agencies’ and ministries’ websites up until 30 June 2020, thus, additional response beyond this timeline, as well as responses taken by other government ministries and the private sector (if any), which could potentially be valuable, were beyond the scope of this study. Secondly, data was extracted from publicly available domains but we did not conduct any verification and triangulation on each action publicly reported by the relevant agencies. Additionally, other measures taken but not documented publicly, could not be captured with the study’s search strategy.

### 4.4. Future Studies

As the document review approach could not detect the measures that were not publicly documented, to facilitate comprehensiveness of country sharing, future studies should explore public health measures and strategies taken within the ministry that were not made public, as well as the numerous and myriad responses taken by the private health sector and third sector organisations.

## 5. Conclusions

Malaysia’s response to COVID-19 began before the detection of the disease in the country and intensified when evidence of local spread emerged. Countrywide and localised lockdowns, targeted screening approach and quarantine systems involving whole nation collaboration were employed to combat the pandemic, with the provision of equal care regardless of citizen and social status. Malaysia’s responses addressed major actions called for by WHO and appeared to be able to contain and mitigate the country’s COVID-19 outbreak by the end of June.

## Figures and Tables

**Figure 1 ijerph-18-11109-f001:**
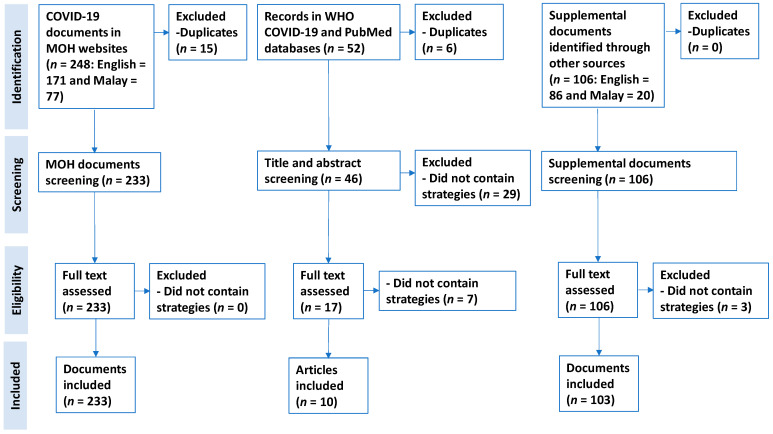
PRISMA flow diagram illustrating document search and selection.

**Figure 2 ijerph-18-11109-f002:**
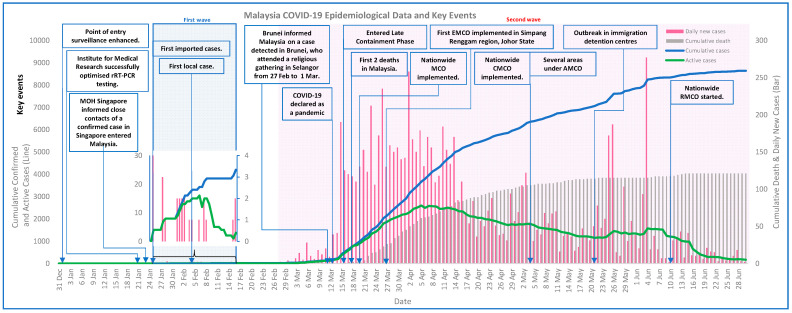
Key events and reported COVID-19 cases in Malaysia.

**Figure 3 ijerph-18-11109-f003:**
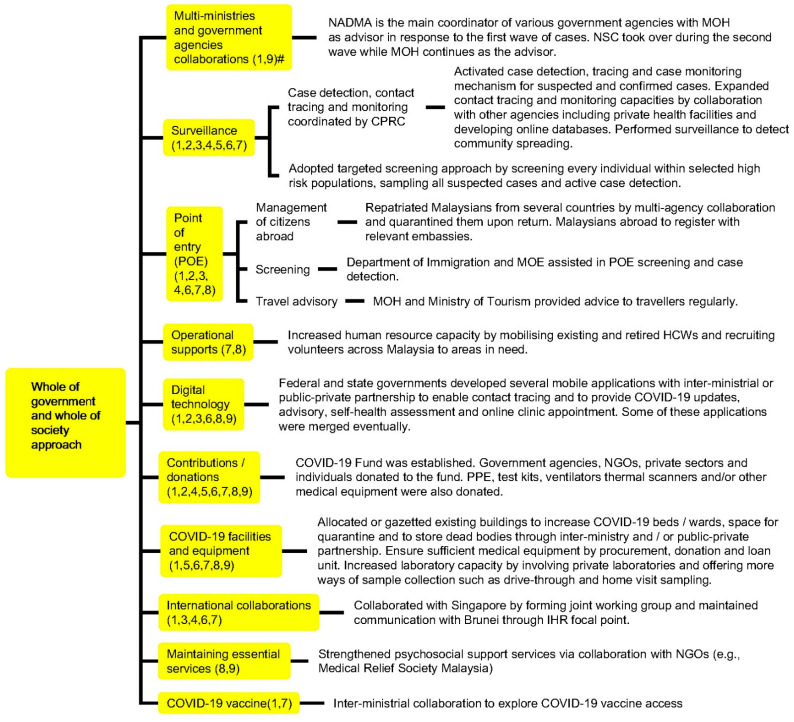
Whole of government and whole of society approach taken by Malaysia in response to the COVID-19 pandemic. # Numbers in parentheses denote WHO Pillars.

**Figure 4 ijerph-18-11109-f004:**
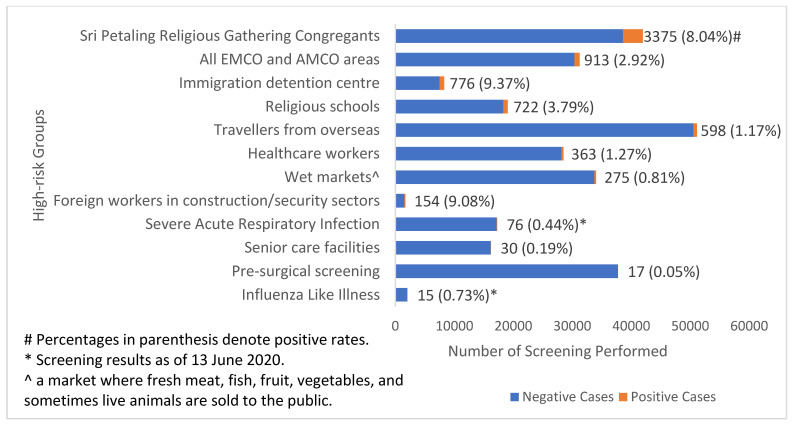
Targeted screening results for 12 identified high-risk groups as of 19 June 2020.

**Figure 5 ijerph-18-11109-f005:**
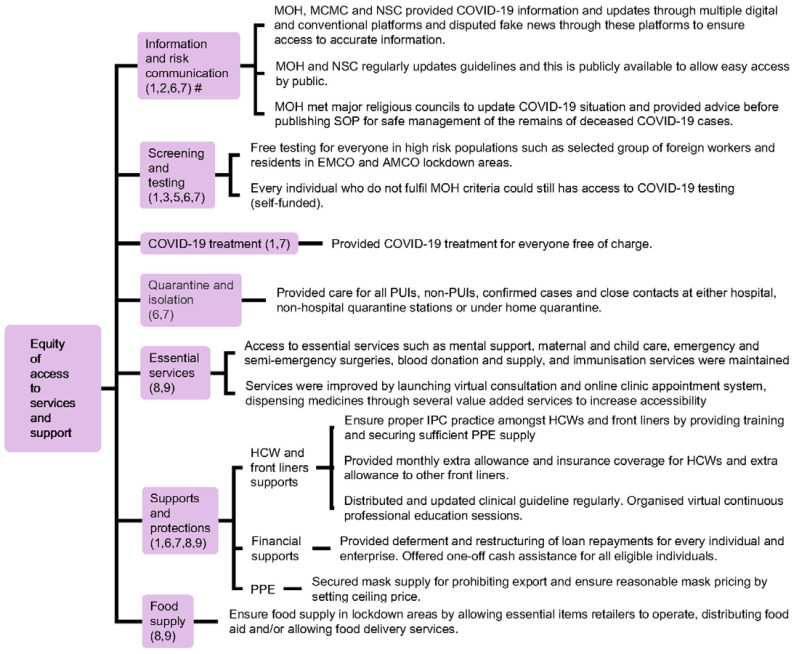
Equity of access to services and support provided by the Malaysian government during the COVID-19 pandemic. # Numbers in parentheses denote WHO Pillars.

**Figure 6 ijerph-18-11109-f006:**
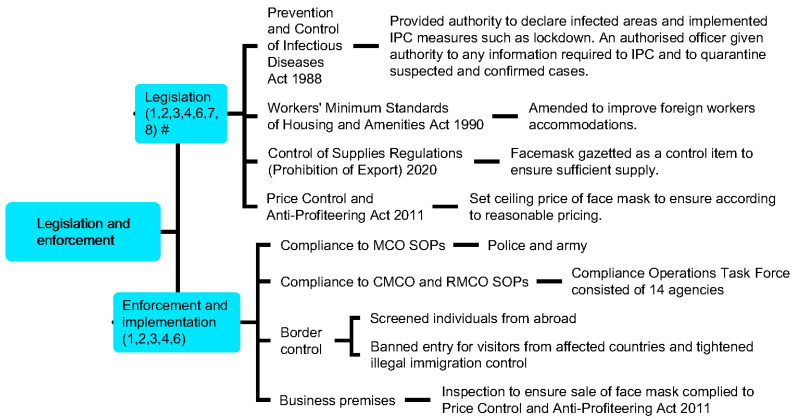
Relevant laws and regulations enforced by the Malaysia in response to the COVID-19 pandemic. # Numbers in parentheses denote WHO Pillars.

**Table 1 ijerph-18-11109-t001:** Types of cordon sanitaire/lockdowns implemented by Malaysia.

Types of Lockdown	Nationwide Partial Lockdown			Localised Partial Lockdown with Mass Screening	Localised Complete Lockdown with Mass Screening
Name	Movement Control Order	Conditional Movement Control Order	Recovery Movement Control Order	Administrative Movement Control Order	Enhanced Movement Control Order
Period of time	18 March–4 May	5 May–9 June	10 Jun–31 August	2–5 weeks until screening completed	
Reason(s) of implementation	Second wave of outbreak spread nationwide	Fulfilled 6 WHO criteria to lift lockdown	Local transmission controlled	Localities with sudden increase of cases	Localities with sudden and continual surge of cases
Movement restriction	Highly regulated restriction of movement	Interstate travel allowed with permission	No restriction	Allowed movement within the localities	Highly regulated restriction of movement
Socioeconomic activities	All closed except essential services	(i) Allowed most sectors to open subject to SOPs (ii) Allowed sports with <10 people subject to SOPs	(i) Reopened in stages subject to SOPs (ii) Prohibited sports with involving mass gathering	All closed except retailers for essential items and/or food delivery services allowed to operate	All closed. Food was provided, retailers for essential items and/or food delivery services allowed to operate
Mass gatherings	Prohibited	Prohibited except festival celebrations allowed for immediate family members	Religious gathering and festival celebrations allowed with conditions	Prohibited	Prohibited
Points of entry	(i) Citizens not allowed to leave except with permission (ii) Non-citizens not allowed to enter except with permission (iii) 14-day quarantine for those who entered Malaysia			Allowed to enter or leave the area with valid reasons	Not allowed to enter or leave the area except authorised personnel
Educational institutions	Closed	Closed	Reopened by stages	Closed	Closed
Surveillance approach	Targeted screening approach			(i) Targeted screening approach (ii) Set up medical base to screen all residents and collected samples for testing (iii) Conducted house-to-house active case detection.	
Enforcement agency	Police and army	Compliance Operations Task Force		(i) Police, army, Malaysia Civil Defence Force and The People’s Volunteer Corps (ii) Performed disinfection activities	
WHO SPRP pillars	Measures implemented in all lockdowns straddled crossed Pillars 1–9				

**Table 2 ijerph-18-11109-t002:** Summary of Malaysia’s COVID-19 quarantine and isolation systems based on MOH guidelines.

Quarantine Stations	Institutionalised Quarantine Systems		Home Quarantine
	Hospital Isolation Room	Non-Hospital—Low Risk Quarantine and Treatment Centres	
Facility requirements	Single room (compulsory for PUIs) preferably with closed door and en-suite bathroom	(i) Preferably single room with good ventilation and with beds at least 1 m apart and equipped with en-suite bathroom (ii) With room for clinical examination and store room for PPE, medications, consumables, and linen (iii) Repurposed from existing buildings with multi-agency collaboration	(i) Single bedroom with en-suite bathroom or frequently disinfected common bathroom (ii) Accessible to food, disinfectant, gloves, facemasks, and other daily essentials (iii) Able to obey instruction, practise social distances with high-risk family members (e.g., elderly), and seek medical attention with own transportation
Criteria for admission	(i) Before 18 Mar: all PUIs and confirmed cases (ii) After 18 Mar: (a) PUIs—clinically ill, uncontrolled medical conditions, immunocompromised, pregnant, <2 years old or >60 years old (b) Confirmed cases—mild symptoms or asymptomatic (c) Symptomatic close contacts	After 18 Mar: (i) Confirmed cases who were asymptomatic or mildly symptomatic (ii) PUIs who were unable to do home quarantine (iii) Confirmed cases who fulfilled criteria for step-down facility (iv) 3 April–9 June: Individuals from abroad	(i) Before 18 March: non-PUIs from affected countries (ii) After 18 March: outpatient PUIs could fulfil above criteria and discharged PUIs (iii) 18 March–2 April: any individual from aboard who was not a PUI (iv) After 10 June: asymptomatic flight/ship passenger and crew with negative COVID-19 testing
WHO SPRP pillars	4,6,7,8	1,4,6,7,8	4,6,7,8

## Data Availability

All data generated or analysed during this study are included in this published article and available through the corresponding author upon request.

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
