# Peer review of "Malaysia’s Health Systems Response to COVID-19"

_ijerph, 2021, doi:10.3390/ijerph182111109_

Round 1

Reviewer 1 Report

A paper of high quality, well documented. Great presentation, use of language. Tables and figures are really helpful and support the data used.

Reviewer 2 Report

General comments.

Some figures have terrible quality. Either re-save them in higher PDI (e.g. 600 DPI)  or save them as vector graphics.

Line 14: what is “mapped with domain”?  Aligned or synchronized? Please clarify or rewrite.

Line 22: It is not clear for non-Malaysian readers what is the difference in treatments based on citizenship. Please clarity here.

Line 37; R0 is only responsible for transmission but not for the fatality rate. Here you are referring to high fatality rate. Based on what?  Either: talk only about transmissivity or if you are talking about high fatality please provide a reference. You ref talks only about R0 and not about fatalities.

Line 72: Can you provide a link for the database? Can you provide a ref for the WHO database address. Did you just look on the web site? If so please clarify.

Line 74-76 Sentence too long. Please split your thought in two.

Line 78-80: Please put those keywords into a table.

Line 82: Did you include non-peer reviewed pre-prints or not in your review. Please be more specific about the inclusion criteria.

Line 84: Can you provide the links for the sources of those sites where you eventually took the information from?

Line 86-88: Can you be more specific about the inclusion criteria? Did you search for your keyword? If found what exactly (at minimum) should be in the article to be included. What do you mean by “any measures”? Please be specific. Did you look only at the abstract?

Line 92: Did you verify only 10% of those originally selected?  Please clarify! Why only 10%?

Line 93-94: Did the second reviewer verified only 10% here as well?

Line 100: How did you define subdomains? Can you please provide the details on the process?

Lines 102-103: did you include only those matched by all five? Is that what was verified?

Line 111-113: Were those MOH all in a report formats or some were just announcements? How do you define a document here?  Can you also specify how many were in English and in Malay? Say with WHO. What do you mean exactly by document here? Is that a pdf report? Paper? Notice?

Line 118: “cordon sanitaire”. Is that in French? Does that mean entry ban? Please clarify.

Figure 2: What is IMR? Can you explain in caption? Please explain how Selangor is related to Brunei geographically.

Line 149: Why religion schools were at high risk? Please provide the details or ref!

Line 154: Why this region got some much attention? Please provide a background or reference(s).

Line 162: Reference for that?

Figure 03: Can you clarify precisely what you mean by wet markets? It can mean different things.

Figure 04: Quality is terrible. Either use a vector graphic or a higher DPI (400 and up I guess) image.

Reviewer 3 Report

Dear Authors,

the manuscript has well written and interesting, even if is the structure is principally a report.

I suggest to improve the captions of figures, such as the caption of figure 2; the figure, in fact,  is complicated while the caption synthetic.

Best Regards

Reviewer 4 Report

Comments

Title: Malaysia’s health systems response to COVID-19.

This study aimed to highlight COVID-19 response by the Ministry of Health (MOH) and the Government of Malaysia in order to share Malaysia’s lessons and to improve future pandemic preparedness. The study methodology relied on using a qualitative approach known as a rapid review methodology. This was performed to identify publicly available information on the measures employed by Malaysia to address the COVID-19 pandemic via different Ministry of Health webpages and other online databases.

The presentation of article was generally good with some interesting points made. The discussion and conclusion support the literature presented, with some recommendations for future research having been made.
